# Motor planning flexibly optimizes performance under uncertainty about task goals

Aaron L. Wong[1] & Adrian M. Haith[1]

In an environment full of potential goals, how does the brain determine which movement to execute? Existing theories posit that the motor system prepares for all potential goals by generating several motor plans in parallel. One major line of evidence for such theories is that presenting two competing goals often results in a movement intermediate between them. These intermediate movements are thought to reflect an unintentional averaging of the competing plans. However, normative theories suggest instead that intermediate movements might actually be deliberate, generated because they improve task performance over a random guessing strategy. To test this hypothesis, we vary the benefit of making an intermediate movement by changing movement speed. We find that participants generate intermediate movements only at (slower) speeds where they measurably improve performance. Our findings support the normative view that the motor system selects only a single, flexible motor plan, optimized for uncertain goals.

[1] Department of Neurology, Johns Hopkins University School of Medicine, Baltimore, Maryland 21287, USA. Correspondence and requests for materials should be addressed to A.L.W. (email: aaron.wong@jhu.edu).

In daily activities, we are often forced to act before we know with complete certainty what the goal of our movement will be. For example, a goalkeeper in a soccer game might need to decide in which direction to move before it is clear where the ball is going to go. In laboratory-based tasks when people are faced with similar uncertainty about which goals to pursue, they frequently generate movements that are intermediate between the potential goal locations[1–12].

These intermediate movements are thought to provide fundamental insight into how we prepare actions to achieve desired goals. The prevailing explanation for the occurrence of intermediate movements is that they represent an averaging of the distinct motor plans that have been prepared for each potential motor goal (for example, to each of the presented targets on a screen)[13,14]. As such, the occurrence of intermediate movements has often been viewed as strong evidence that multiple motor plans are developed by the motor system in parallel for each potential goal, and that these plans become unintentionally blended into a single action outcome through movement averaging[9,10,12].

An alternative theory of intermediate movements proposes that—rather than being accidental—such movements are actually purposeful, reflecting a single, deliberate motor plan that seeks to optimize performance under goal uncertainty[11,15]. If there are two potential goals for a reach, employing an intermediate movement allows an individual to initially bring their hand closer to both targets, affording more time to determine the true goal location before making a commitment to either one. Thus, according to this normative theory, the motor system considers multiple goals in parallel, but generates only a single, purposeful movement plan that is deliberately directed between the two goals.

A critical prediction of this normative theory is that intermediate movements should only occur when they are advantageous. For instance, if two potential targets are close together, an intermediate movement is useful as it brings the hand close to both targets. The same is not true, however, for targets that are widely separated[11]. Consistent with this theory, intermediate movements are rarely observed for targets spaced far apart[11,16]. However, from the point of view of movement averaging, this could be attributable to the fact that motor plans for closely separated targets are likely to be quite similar and therefore afford a greater likelihood of being averaged[7,13,14]. Thus on the strength of current evidence, it is difficult to dissociate between these two theories[11].

A potentially more decisive prediction of the normative planning theory is that the occurrence of intermediate movements should be modulated by movement speed. When moving slowly, more time is available to alter a movement in light of new information acquired after it has started. An intermediate-movement strategy is therefore more advantageous for slow compared to fast movements; intermediate movements should occur less frequently when participants move quickly. In contrast, the movement averaging theory does not predict an effect of movement speed or success rate on the likelihood of producing an intermediate movement. Therefore, to distinguish between these two hypotheses, we tested how human participants behaved while reaching to an ambiguous target array at varying speeds. We find that participants generate intermediate movements only at slower speeds, where they lead to measurably better task performance. This supports the normative view that intermediate movements arise from a single deliberate motor plan that is optimized to address uncertain goals.

## Results

### Behavioural paradigm for movement planning.
Participants performed reaching movements under a 'go before you know'

paradigm[6] in which they were required to initiate a movement towards either one (single-target trials) or two (dual-target trials) potential targets (Fig. 1a; see Methods for details). Targets could be located at one of four eccentricities from the midline (for dual-target trials, pairs of targets appeared symmetric about the midline; Fig. 1b). In dual-target trials the true location of the target was not disclosed until after the hand began moving towards the targets—a reliable means of eliciting intermediate movements[6,10,15]. Critically, participants were instructed to perform these reaches at either 'Slow' (peak velocity, 0.3–0.7 m s$^{-1}$, corresponding to an average movement time of $0.61 \pm 0.10$ s) or 'Fast' (peak velocity, 0.8–1.5 m s$^{-1}$, corresponding to an average movement time of $0.33 \pm 0.02$ s) movement speeds (Fig. 1c). We measured participants' initial reach errors to determine where movements were initially directed before the correct target was known (see Methods for details). If intermediate movements reflect a purposeful strategy, participants should generate them more often when moving slowly compared to moving quickly, and this effect might also depend on the separation angle between potential targets.

### Proportion of intermediate movements depended on speed.
On dual-target trials, we observed significant differences in participants' behaviour depending on their instructed movement speed. Trajectories from a representative participant are shown in Fig. 1d (movements from all 16 participants in Experiment 1 are shown in Supplementary Fig. 1). When moving slowly, this participant tended to reach in a direction intermediate between the two targets, as expected based on prior studies[1,3,5–7,9–12]. In contrast, when moving quickly, this participant almost exclusively reached directly towards one of the two targets. Across all participants, the distributions of initial reach errors (Fig. 2) were clearly different between Slow and Fast movements at all target-separation angles (Kolmogorov-Smirnov test on pooled data, $p < 0.001$ for each target-separation angle).

Importantly, this effect of movement speed was not attributable to differing amounts of preparation time taken for different movements. Although reaction times were longer for Slow compared to Fast trials (Slow: $255.47 \pm 11.95$ ms, Fast: $215.21 \pm 8.15$ ms; paired $t$-test, $t(15) = 3.25$, $p = 0.005$) there was no correlation between reaction time and normalized aiming direction from the midline (see Methods; Slow: $r^2 = 0.01$, Fast: $r^2 = 0.003$). That is, intermediate movements were performed at comparable reaction times to movements aimed directly at one of the two targets.

To more precisely quantify the likelihood of a participant generating an intermediate movement at different speeds, we fit a probabilistic mixture model to the distribution of initial reach errors (Fig. 3a; see Methods for details). This analysis confirmed that participants were less likely to generate intermediate movements when they moved quickly compared to slowly (significant effect of instructed speed: $\chi^2(1) = 50.54$, $p < 0.001$; Fig. 3b) and when the targets were separated more widely ($\chi^2(1) = 4.02$, $p = 0.04$; the coefficient of this main effect was significantly less than zero indicating that participants were less likely to generate an intermediate movement as target-separation angle increased, bootstrap analysis, $p = 0.045$). There was no significant interaction between instructed speed and target separation ($\chi^2(1) = 0.00$, $p = 0.96$).

In principle, the effect of speed on intermediate movements might have been due to participants for some reason adopting qualitatively different behaviours in response to the differing speed instructions. However, spontaneous variations in movement speed within each instructed-speed condition affected the likelihood of generating an intermediate movement;

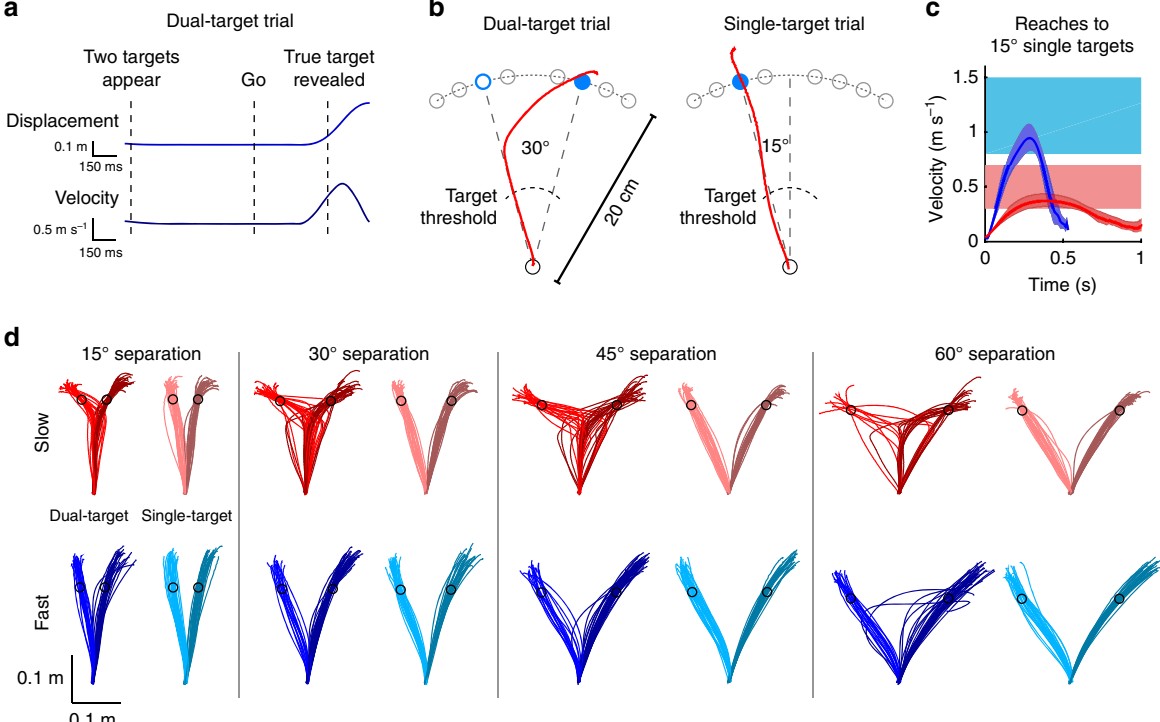

**Figure 1 | Experimental paradigm and example data from one participant.** (**a**) In dual-target trials, participants observed two potential target goals. The true target was revealed only after a reaching movement was initiated (that is, when the hand moved 25% of the distance to the targets). (**b**) Example trajectories for dual- and single-target trials. The filled blue circle indicates the true target location, the open blue circle indicates the other potential target; open grey circles represent locations where targets may appear in other trials. The black dashed line indicates the invisible threshold (25% of the distance to the targets) at which the correct target was revealed. (**c**) Average velocity profiles for a representative participant. Peak speed for each movement was required to fall within one of two specific ranges, depending on the condition (Slow or Fast). Error bars are s.d. (**d**) Data from the same participant showing reach trajectories at Slow and Fast movement speeds. Reaches were made under either dual-target (left traces in each panel) or single-target (right traces in each panel) conditions. Within a given set of movements, the darker-shaded reaches are those that ultimately went to the rightward target.

movements that had a slow speed early in the reach were more likely to be intermediate compared to faster movements (Fig. 4). For all target-separation angles, the proportion of intermediate movements was significantly greater for the slowest quartile compared to the fastest quartile of reaches within each instructed speed: the proportion of intermediate movements ($\theta$ in the mixture model; see Methods) was greater by at least 0.16 for slower reaches compared to faster reaches within both the Fast and Slow conditions (non-overlapping 95% confidence intervals). The differences in intermediate-movement behaviour were therefore not purely a consequence of the differing instructions ('Slow' or 'Fast') provided to participants, but were a direct consequence of the speed of the movement.

**Selection of reach strategy optimized task performance.** The core premise of our speed manipulation was that the benefit of adopting an intermediate movement strategy should depend on movement speed. The fact that participants exhibited both intermediate and direct movement strategies at a range of different speeds enabled us to validate that this was the case (Fig. 5). At slower initial reaching speeds (0.20–0.30 m s$^{-1}$), participants achieved better success rates (that is, intercepting the correct target while maintaining a movement speed within the required instructed speed criterion) for intermediate movements than for direct reaches: intermediate reaches had a 74.5% success rate, compared with 48.3% for direct reaches. At faster speeds (0.40–0.50 m s$^{-1}$), however, this pattern was reversed; intermediate reaches had a 34.9% success rate while direct reaches had a

44.7% success rate. Critically, the strategy that participants favoured at different speeds was consistent with these contrasting success rates; participants generated intermediate movements 75.6% of the time at slower speeds, when this strategy yielded better success rates, compared with 15.7% of the time at faster speeds when an intermediate-movement strategy was less successful.

**Reaction time did not influence choice of movement strategy.** One potential concern with our first experiment was that participants were allowed a long time (~700 ms before the go cue) to prepare their movements. Participants may have been transiently susceptible to movement averaging, but, given time, were able to override this default response with a decision to aim directly for one of the two targets. Consistent with this idea, other authors have shown that movement averaging can occur at low RTs (for example, ref. 6). In a second experiment (Experiment 2), we therefore reduced the time available to observe the target locations by synchronizing the go tone with the time at which the targets were presented and requiring participants to initiate their movements within 400 ms (Fig. 6). Under these requirements, participants exhibited similar reaction times for both Slow and Fast trials (Slow: 252.75 ± 6.71 ms; Fast: 262.21 ± 5.52 ms; paired *t*-test, *t*(9) = − 1.59, *p* = 0.15), with no clear distinction between the distributions of RTs for intermediate or direct reaches for any given participant.

Despite dramatically reducing the time available to view the target locations before moving, we observed the same patterns of behaviour as in Experiment 1. Participants exhibited more

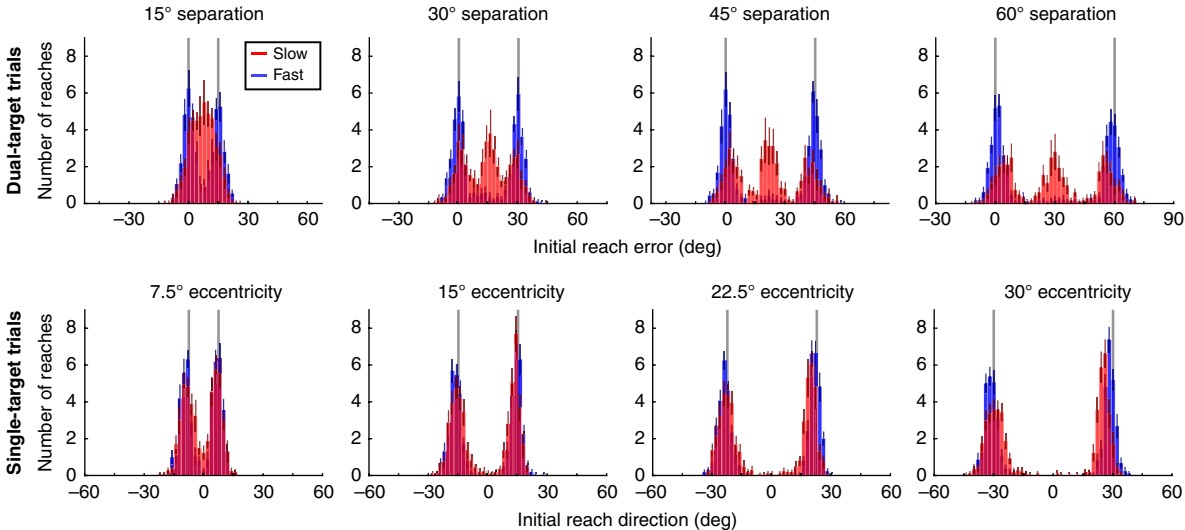

**Figure 2 | Group average of intermediate reach distributions.** Initial reach error distributions in dual-target trials (upper panels) and reaching direction distributions in single-target trials (lower panels) were averaged across all participants ($n = 16$). Data are shown for each target-separation angle (columns) at Fast (blue) and Slow (red) movement speeds. Coloured bars (blue or red) denote s.e.m.; grey lines indicate the position of the targets in each condition. For single-target trials, positive reaches are directed to the rightward target.

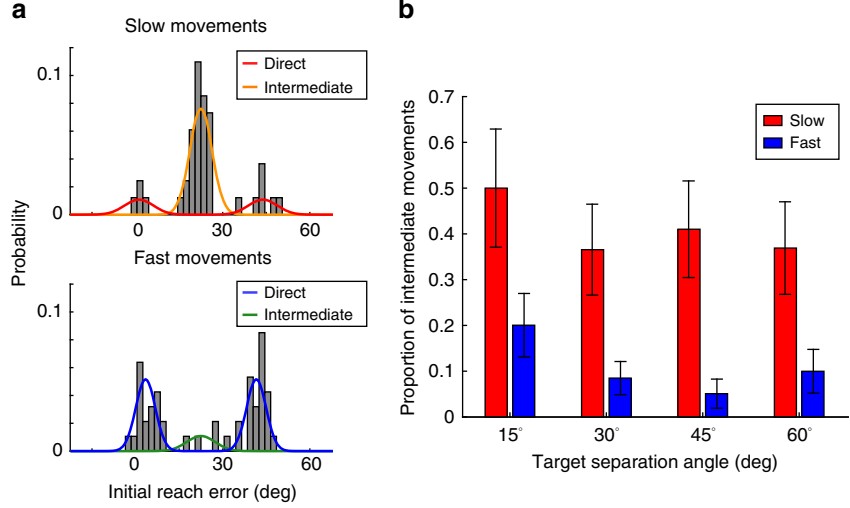

**Figure 3 | Estimated proportion of intermediate movements.** (**a**) Example maximum-likelihood fits to the distributions of Fast and Slow movements for a single participant, for a target-separation angle of 45°. Grey histograms indicate the initial-reach-error distribution. Smooth lines indicate direct (bimodal) and intermediate (unimodal) components of the fitted model. (**b**) Average proportion of intermediate movements ($\theta$) at different movement speeds and target-separation angles across all participants ($n = 16$). Error bars represent s.e.m.

intermediate movements in Slow blocks than in Fast blocks (proportion of intermediate movements: Slow, $0.21 \pm 0.07$; Fast, $0.03 \pm 0.01$; significant effect of instructed speed: one-tailed paired $t$-test, $t(9) = 2.15$, $p = 0.03$). The observed proportion of intermediate movements in Experiment 2 was not significantly different from that in Experiment 1 (non-significant difference between experiments, $\chi^2(1) = 1.66$, $p = 0.20$); to the extent that there were any differences between the experiments, reducing the available preparation time led to fewer intermediate movements. If movement averaging arose from a more automatic process, this would have predicted the opposite result.

Other key results from Experiment 1 were reproduced in Experiment 2. The frequency of intermediate movements depended on initial movement speed within each condition, for both Slow and Fast reaches (non-overlapping 95% confidence intervals between lower and upper quartiles for both Slow and Fast movements). Furthermore, the success rate of each movement type

corresponded to the likelihood of it being employed: for slower reaches ($0.28$-$0.30\,\mathrm{m\,s}^{-1}$), intermediate movements had a greater success rate compared to direct reaches (intermediate, 55% correct; direct, 27% correct) and correspondingly tended to be more prevalent (57% of reaches were intermediate), whereas the opposite was true at faster ($0.36$-$0.38\,\mathrm{m\,s}^{-1}$) speeds (intermediate, 27% correct; direct, 40% correct; 31% of reaches were intermediate). Thus, the amount of time available prior to movement onset did not change the main finding that participants transitioned between intermediate and direct reaches depending on the relative benefit of these actions at a particular movement speed.

**Speed affected the direction of reaches to single targets**. In addition to the effects of movement speed on intermediate movements in dual-target trials, we also found effects of movement speed on behaviour in single-target trials. On Slow

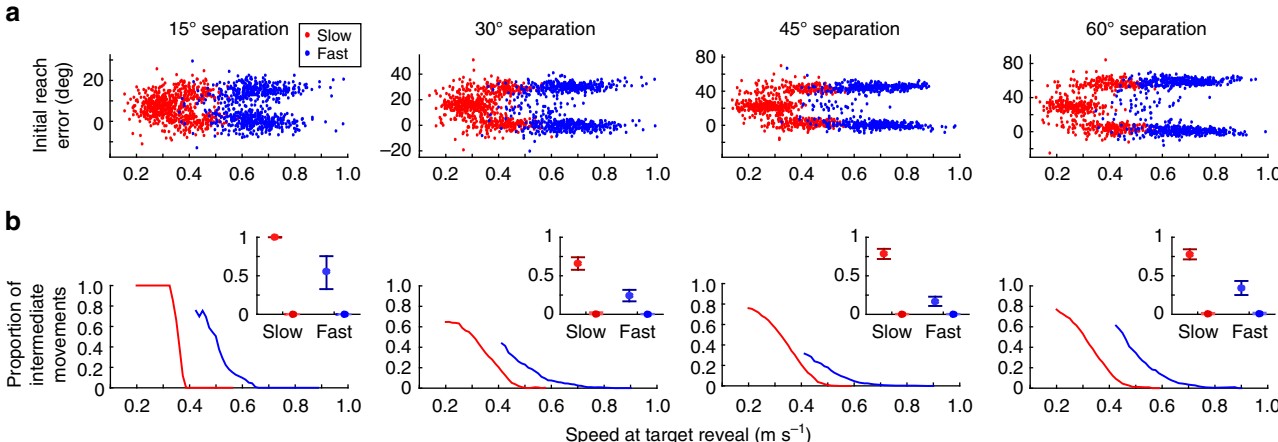

**Figure 4 | Relationship between early reach velocity and proportion of intermediate movements.** Data are pooled across all participants (n = 16). (**a**) Initial reach errors were plotted against reach speed at 4 cm (that is, the time when the correct target was revealed) for reaches in the instructed-Slow (red) and instructed-Fast (blue) conditions. Individual panels are different target-separation angles. (**b**) Proportion of intermediate movements for a given early reach velocity. Inset, estimated value of the proportion of intermediate movements with bootstrapped 95% confidence intervals, for instructed-Slow and instructed-Fast reaches. Darker confidence intervals (at left) represent reaches from the lower quartile of the instructed speed range, and lighter confidence intervals (at right) represent reaches from the upper quartile of the instructed speed range.

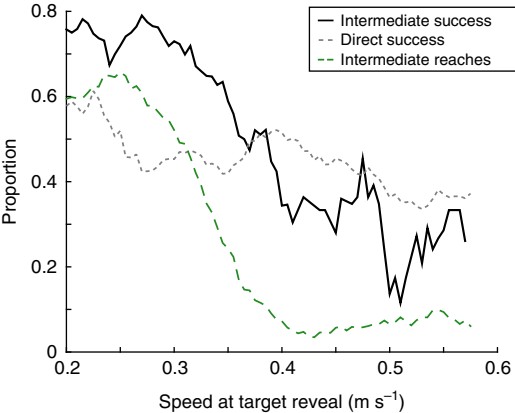

**Figure 5 | Success rate and proportion of intermediate movements generated as a function of movement speed.** For intermediate (black) or direct (grey) reaches, the proportion of successful trials pooled across all participants (n = 16) was estimated for each reaching speed measured at 4 cm. Additionally, the proportion of trials assigned a classification of intermediate (green) for the purposes of computing success rate are shown for comparison.

single-target trials in Experiment 1, we found that movement direction was consistently biased towards the centre of the array of potential target locations (Fig. 7). The magnitude of this bias increased as a function of target eccentricity (one-way ANOVA, $F(3,15) = 8.85$, $p < 0.001$), consistent with previous findings[17–19]. Note, because there were no 'intermediate' reaches generated on single-target trials, these biases reflect a tendency to aim slightly inward towards the midline on Slow trials (Fig. 2b). This effect depended on experience performing the task; reach-direction biases in single-target trials performed in the very first training block (when participants were learning to reach at the correct movement speed) did not differ across target eccentricity (no effect of target-separation angle in training blocks, $\chi^2(1) = 0.95$, $p = 0.33$; significant interaction between target eccentricity and whether the movement was generated during the initial training block or during the test blocks; $\chi^2(1) = 6.15$, $p = 0.01$).

This use-dependent bias in movement direction occurred only for Slow movements. For Fast movements, initial reach directions

showed little dependence on target eccentricity (one-way ANOVA, $F(3,15) = 2.55$, $p = 0.06$; a two-way ANOVA revealed a main effect of instructed speed ($F(1,15) = 44.99$, $p < 0.001$) as well as a significant interaction between speed and target eccentricity ($F(3,15) = 2.81$, $p = 0.04$).

The same pattern of movement biases observed in single-target trials was also apparent for non-intermediate (direct) movements in dual-target trials ($\mu_d$ in the mixture model; two-way ANOVA, main effect of speed, $F(1,15) = 70.81$, $P < 0.001$ and a marginally significant interaction between speed and eccentricity, $F(3,15) = 2.30$, $P = 0.08$). The bias was slightly larger on dual-target trials compared to single-target trials ($\chi^2(1) = 23.30$, $P < 0.01$; Fig. 7). On average, however, the magnitude of this difference in bias was quite small (Slow, 1.15°; Fast, 0.35°). Experiment 2 similarly revealed a difference in bias between Slow and Fast reaches for both single-target (Slow: 2.23° ± 0.98°; Fast: −0.16° ± 0.31°; paired t-test, $t(9) = −2.25$, $P = 0.05$) and dual-target trials (Slow: 2.84° ± 0.72°; Fast: 0.17° ± 0.25°; paired t-test, $t(9) = −3.83$, $P < 0.01$). While part of this measured bias in dual-target trials in both experiments may arise from misclassification of intermediate movements as direct, the effect of this confound should diminish for larger target separations, although we observed the opposite trend. Therefore, the tendency for direct movements to be biased towards the midline appeared to be similar across both single and dual-target trials. These similarities in behaviour across trial types existed despite significant differences in reaction times in Experiment 1 (single-target: 220.06 ± 8.15 ms, dual-target: 235.34 ± 8.14 ms; paired t-test, $t(15) = 2.99$, $P = 0.01$); there was no difference in reaction time in Experiment 2 (single-target: 257.06 ± 5.37 ms, dual-target: 257.90 ± 5.61 ms; paired t-test, $t(9) = −0.36$, $P = 0.73$).

Interestingly, this speed-dependent bias did not appear to be a consequence of generating prior intermediate movements: in Experiment 1, for example, three participants did not generate any intermediate movements throughout the experiment, yet still showed an inward reach-direction bias on Slow compared to Fast reaches on single-target trials (two-way ANOVA, main effect of speed, $F(1,3) = 60.15$, $p < 0.001$) and on dual-target trials (two-way ANOVA, main effect of speed, $F(1,3) = 45.84$, $p < 0.001$). Thus, this speed-dependent bias was not dependent on the specific types of movements (direct or intermediate) that participants made on dual-target trials.

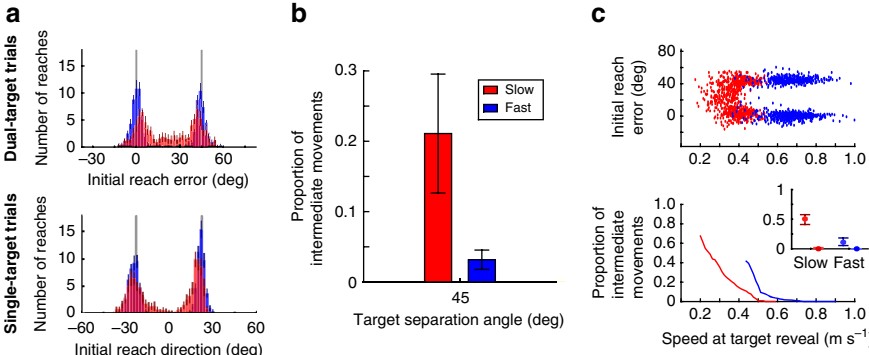

**Figure 6 | Results from Experiment 2.** (**a**) Distributions of average initial reach errors in dual-target and single-target trials for all participants ($n = 10$), as in Fig. 2. Data are shown for Fast (blue) and Slow (red) movement speeds. Grey lines indicate the position of the targets in each condition; for single-target trials, positive reaches are directed to the rightward target. (**b**) Average proportion of intermediate movements as estimated from the mixture-model fits. (**c**) Relationship between early reach velocity and proportion of intermediate movements, as in Fig. 4. Top panel, initial reach errors plotted against reach velocity at 4 cm for Slow (red) and Fast (blue) conditions. Bottom panel, proportion of intermediate movements estimated at a given initial reach velocity. Inset, estimated value of the proportion of intermediate movements with bootstrapped 95% confidence intervals, for the lower (darker confidence intervals) and upper (lighter confidence intervals) quartiles in each of the Slow and Fast conditions.

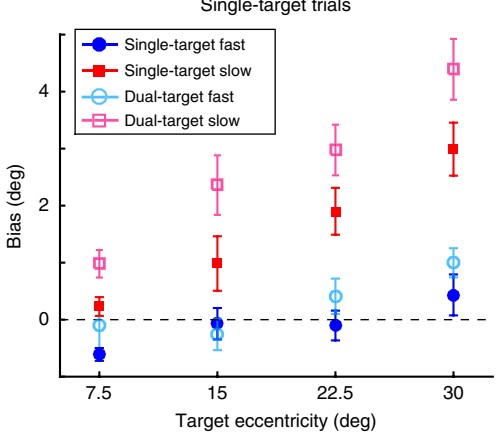

**Figure 7 | Average inward bias in initial reach direction for Experiment 1 as a function of target eccentricity.** Data were averaged across all participants ($n = 16$) for single-target biases (Slow movements, red; Fast movements, blue) and dual-target biases based on $\mu_d$ from the mixture-model fits (Slow movements, pink; Fast movements, cyan). Positive values indicate direct reaches were biased towards the midline and negative values indicate reaches biased away from the midline; error bars represent s.e.m.

## Discussion

Our findings reveal that the occurrence of intermediate movements in the presence of multiple potential motor goals is strongly affected by movement speed as well as the separation between potential targets. This change in behaviour with movement speed mirrored a change in the success rate of intermediate movements relative to direct movements, suggesting that intermediate movements are produced as a purposeful strategy to maximize task success[11,15].

Intermediate movements are commonly thought to occur because the motor system prepares more than one plan in parallel, which then become averaged together. Indeed, the fact that intermediate movements occur at all is often taken as evidence for the existence of parallel plans[6,9,10,12]. Such conclusions, however, overlook the critical alternative explanation that intermediate movements represent a deliberate and purposeful strategy to optimize performance[11,15]. It is generally difficult to dissociate movement averaging from

optimization because the optimal course of action is often indistinguishable from an average of the set of single-target motor plans (or feedback control policies[20]).

One example where averaging and optimization appear potentially dissociable is a study by Stewart and colleagues[10], who used a similar dual-target paradigm to show that if an obstacle blocked a direct movement to one of the targets, intermediate movements become biased away from the obstacle. This occurred despite the fact that the unshifted intermediate movement would not have been impeded by the obstacle. The authors concluded that participants must have been averaging their motor plans for each target. However, this bias could also be explained within an optimization framework: the obstructed target may be associated with a greater cost (that is, reduced value) since it requires more effort to avoid the obstacle and bears the risk of a penalty for an accidental collision. Indeed, the magnitude of excursion around an obstacle depends directly on the amount of punishment associated with hitting it[21,22]. Thus, the asymmetric value of the obstructed and unobstructed targets could account for the biasing of intermediate movements towards the unobstructed target in a similar manner to how intermediate movements are biased in response to asymmetric target probabilities[6], unequal salience[23], uncertainty[24], reward prospect[25] or intent to deceive[26].

Movement averaging has also been suggested at the level of feedback control policies. The gain of feedback responses during movement is known to be modulated by target width[27,28], with narrow targets leading to high feedback gains compared to wide targets. Gallivan and colleagues[20] claimed that ambiguity about the width of a target gives rise to averaged feedback gains, suggesting that feedback control policies are specified in parallel. However, such a pattern of behaviour is also consistent with participants simply preparing a single control policy that balances the effort costs associated with a high feedback gain against the probability that this greater effort will be necessary: participants may adopt an intermediate gain in the knowledge that it will conserve effort while allowing them to be successful most of the time.

These and similar findings that purportedly support movement averaging can all plausibly be explained in terms of participants selecting a single, optimized plan. Conversely, there are many observations that are consistent with the optimization theory, but are difficult to explain in terms of movement averaging. Although optimal behaviour often resembles an averaged movement, there are instances where the optimal solution is to *not* average.

Moving quickly, as we tested here, is one example. Another example is to simply place a virtual barrier in between potential goals to penalize intermediate movements[11]. In this case, participants easily stop generating intermediate movements and instead move directly to one target or the other. These results demonstrate that intermediate movements are not obligatory, and are therefore unlikely to arise from a low-level movement-averaging process.

The flexibility of motor planning is also apparent in our present data through the significant variability of the likelihood of producing intermediate movements both within and across participants. For example, a given participant typically generated a mixture of intermediate and direct reaches for the same movement speed; moreover, the probability of a movement being intermediate depended in part on task instructions (in Fig. 4, compare the slowest movements in the Fast condition, which were intermediate, with the fastest movements in the slow condition, which were direct). Similarly, several subjects in our experiments generated no intermediate movements in any condition. This heterogeneity across subjects is quite common: in many similar data sets, some subset of individuals do not exhibit any intermediate movements (for example, refs 9,10,12) and are often consequently excluded from further analyses.

Variations in strategy across individuals could have arisen from differing valuations of subjective costs (success versus effort), differing attitudes to risk, uncertainty in the decision about which strategy to use, or even because some participants simply never hit upon an intermediate movement strategy. Regardless, variability both within and across individuals is difficult to reconcile with movement averaging, which would presumably be obligatory. Instead, these observations are entirely consistent with the idea that participants were seeking to optimize their performance, selecting—perhaps stochastically or through trial and error—a single motor plan for each trial.

In addition to behavioural evidence, which we have argued does not support movement averaging, advocates of the movement averaging theory have also appealed to neural data and hypotheses arising from those data. The logic of these arguments is based on the observation that neural representations for multiple targets may simultaneously exist in motor regions of the brain[29]. Specifically, neurons in anterior parts of premotor cortex appear to be tuned to distinct targets during a pre-movement delay period; when two targets are presented simultaneously, neurons tuned to both targets become active[29]. Computational theories have posited that such a population of tuned neurons could give rise to movement averaging[7,13,14,30]. Although mutual inhibition between dissimilarly tuned neurons leads to selection of a single action, mutual excitation between similarly tuned neurons could promote integration of neighbouring plans into a single, averaged plan[7,13,14]. Such theories can account for why the occurrence of intermediate movements declines with increasing target separation.

It is difficult, however, to imagine how such a population-coding scheme could be extended to account for the speed-dependent effects on intermediate movements revealed in our data. Although neurons in motor cortex have been reported to represent movement speed through increased or decreased firing rates during movement[31,32], no systematic effects of movement speed have been observed during the preparatory period[33] when averaging is believed to occur. Additionally, relative differences in neural firing rates during movement preparation are already thought to represent the likelihood of each goal being the correct target[34,35]. It is implausible that target probability and movement speed—two independent parameters—could both be represented by modulations of firing rate. Thus our observation that movement speed modulates intermediate-movement behaviour is difficult to explain within this framework.

The same neural data that gave rise to computational theories of movement averaging have also been interpreted more broadly as evidence of parallel motor planning[14,29,30]. That is, parallel activity associated with multiple targets is thought to represent parallel motor plans prepared for each goal[29]. This interpretation, however, overlooks the subtle distinction between a motor goal and a motor plan. Whereas a plan defines specific details about an impending movement, a goal defines the desired outcome of the movement in more abstract terms. The distinction between a motor goal and a motor plan is analogous to the distinction between a cost function and a control policy in optimal control theory (for more in-depth discussion of this distinction, see refs 36,37). We suggest that the parallel activity identified by Cisek and Kalaska[29] may actually represent an encoding of motor goals, describing the intended action outcomes for which a motor plan will ultimately be specified. More generally, we propose that the motor system entertains multiple goals in parallel, but specifies only a single motor plan.

Although goals and plans are often thought to reside in different regions of the brain (for example, ref. 38), a single brain region (for example, the frontal eye fields) can represent the visual stimulus, the motor goal, and possibly even the motor plan at different times during movement preparation[39–43]. Hence, the implication that multiple parallel motor plans eventually coalesce into a single plan[13,14] could instead simply reflect a transition from competing motor goals to a single, optimal plan.

One potential approach to distinguish plans from goals is through the fact that a plan ought to be predictive of specific details of the upcoming movement. For example, preparatory activity of neuronal populations in posterior portions of dorsal premotor cortex (different from the anterior areas of dorsal premotor cortex in which parallel representations have been observed previously) is highly predictive of specific movement details including kinematics and EMG activity[44]; as such, neurons in these regions of cortex arguably encode a motor plan. However, current theories of how that motor plan is encoded suggest that the instantaneous global state across the entire population represents a single action[45,46]. Thus, dorsal premotor cortex appears to represent motor plans as a single, tuned population of neurons, rather than as a population of tuned neurons, as would be necessary for parallel representation of multiple motor plans.

In summary, in brain areas in which parallel encoding is observed, there is currently no data available to suggest that these regions encode plans rather than goals. Brain regions that do clearly encode motor plans appear to only be capable of supporting a single plan at a time. Thus current neural evidence is consistent with a model in which the motor system entertains multiple goals, but prepares only a single motor plan at any one time.

Surprisingly, our data additionally reveal that movement speed also influenced reach direction in unambiguous, single-target trials: use-dependent biases in initial reach direction[17–19] were eliminated when participants moved quickly. One common explanation for such use-dependent biases is that they arise from a simple Bayesian integration of prior expectations and current observations to estimate the current target location[17]. For example, Verstynen and Sabes[17] demonstrated that reaching to a single target on any given trial is significantly biased by the distribution of previously experienced target locations even for targets as far as 90° away. The authors proposed that these effects arose from a Bayesian integration of current evidence about the target location with prior knowledge about where the target has usually appeared in the past. This Bayesian explanation, however, cannot account for the effect of movement speed that we observed since a bias in the expected goal location should equally affect movements of any speed.

Another potential explanation for the directional bias we observed for direct movements is that they arise through priming[47,48] from intermediate movements generated on dual-target trials (that is, interference between motor plans from one trial to the next), particularly since participants showed single-target biases and intermediate movements primarily in the Slow condition. However, even participants who generated no intermediate movements in our study still exhibited speed-modulated biases towards the midline on both single- and dual-target trials. Hence, these previously proposed theories (priming or Bayesian priors) cannot explain our findings.

Instead, the fact that use-dependent biases and intermediate movements both depend on movement speed suggests that they may derive from the same principle: as a means to manage motor goal uncertainty. That is, use-dependent biases could be viewed as a kind of intermediate movement. A priori uncertainty as to where the target might appear could influence behaviour even after the target has been presented[17]. This uncertainty could lead to a small bias towards the midline on single-target trials, similar to the manner in which two equally probable targets prompt a movement directed midway between them. Indeed, the direction of an intermediate movement is known to be weighted by the degree of asymmetry in goal uncertainty[4,6,9,15], consistent with optimality principles[15]. Thus, use-dependent biases could represent one extreme (that is, near-total certainty about target location) of a continuum of intermediate movements generated under varying degrees of goal uncertainty.

In conclusion, our data provide strong evidence that intermediate movements are not the outcome of involuntary averaging between parallel motor plans, but are generated as the result of a purposeful planning process that flexibly selects a single motor plan to maximize task success. This single-plan theory offers a parsimonious explanation of observed behaviour, is compatible with our current understanding of the neural basis of movement preparation, and provides a coherent conceptual framework for understanding movement planning in the brain.

## Methods

**Participants.** Twenty-seven right-handed, adult (age 18-28, 11 male), neurologically healthy participants were recruited for this study (17 participants in Experiment 1, ten participants in Experiment 2). One participant in Experiment 1 was excluded for not meeting the minimum reach velocity criterion during 'Fast' movement blocks. All participants were naive to the purposes of this study, and provided written informed consent. Methods were approved by the Johns Hopkins School of Medicine Institutional Review Board.

**Experimental paradigm.** Participants made planar reaching movements while their right forearm was placed in a wrist splint and supported by pressurized air jets allowing frictionless elbow and shoulder movements. Vision of the arm was obstructed by a horizontal mirror, through which participants viewed a cursor (0.5 cm diameter) representing the position of the index finger and targets (2 cm diameter) displayed by an LCD monitor (60 Hz) in a veridical horizontal plane. Movement of the index finger was tracked using a Flock of Birds magnetic tracker (Ascension Technology, VT, USA) at 130 Hz. Participants performed a task based on the 'go before you know' paradigm[6]. Two potential targets were presented to the participant, one of which was revealed as the true target location only after the participant started moving. To assess the influence of movement speed on the incidence of intermediate movements in this task, we further required participants to make 'Fast' (peak velocity, 0.8–1.5 m s$^{-1}$) or 'Slow' (peak velocity, 0.3–0.7 m s$^{-1}$) movements during different blocks of the experiment.

In Experiment 1, when participants initiated a trial by moving onto a start target (1.5 cm), potential targets (hollow rings) appeared 20 cm away (Fig. 1b). On dual-target trials (Fig. 1a), two targets were presented symmetrically on either side of the midline at either $+7.5°$ and $-7.5°$, $+15°$ and $-15°$, $+22.5°$ and $-22.5°$, or $+30°$ and $-30°$ (target-separation angles of 15°, 30°, 45° or 60° respectively). On single-target trials, one target appeared at any of the above eight positions. After a random interval (450-1000 ms), an auditory beep cued participants to initiate a movement through the target. The time of movement initiation was determined online as the time when the hand velocity first exceeded 0.1 m s$^{-1}$ or the participant moved 2 cm away from the start target (whichever occurred first).

If movements were initiated more than 500 ms after the go cue, participants heard a buzzer tone and saw a message that they were 'too late'. Once the hand moved 25% (4 cm) of the distance to the target(s), the correct target was filled in to become solid in colour. The trial ended when the participant passed beyond an invisible circle of radius 20 cm centred on the start target. A trial was considered successful if the cursor moved through the correct target circle while satisfying the velocity criterion for that block.

Participants were required to maintain their movement speed throughout the duration of the reach; any attempt to slow down after the target was triggered (that is, a decline of the velocity by more than 40% of the maximum velocity achieved thus far in the trial) led to an immediate termination of the trial. Trials were also cancelled if participants moved very slowly until the target was cued (peak velocity less than 50% of the required minimum velocity threshold by the time of target cuing). Participants were instructed verbally about the speed criterion, and were given visual feedback about their movement speed with a bar whose height reflected peak velocity; a shaded region adjacent to the bar graphically represented the acceptable speed range.

Participants completed ten blocks of trials, grouped into two sets of five blocks. Each set was performed at either a 'Fast' or 'Slow' speed. A set consisted of one training block with only single-target trials to practice the required movement speed (48 trials), followed by four test blocks, each comprising 48 single- and 48 dual-target trials (12 trials at each target-separation angle) randomly intermixed. The order in which the two speeds were performed was counterbalanced across participants.

Experiment 2 was identical to Experiment 1 with two exceptions. The first was that targets only appeared at $\pm 22.5°$ (that is, on dual-target trials this corresponded to a 45° target separation angle); because of this, after each speed-training block (48 trials) only two blocks of test trials (comprised of 48 single- and 48 dual-target trials) were performed at a given speed. Second, following a random inter-trial interval (450–1,000 ms) the target(s) appeared at the same time as the auditory go tone; participants were required to initiate their reach to the target within 400 ms of target onset. These changes were imposed to reduce the time available to deliberate about the stimulus arrangement prior to movement onset, analogous to the timing used by Chapman et al.[6].

**Data analysis.** Reaches were analysed offline using programs written in MATLAB (The MathWorks, Natick, MA, USA). Movement onset was identified according to a velocity criterion (tangential velocity greater than 0.05 m s$^{-1}$), and verified by visual inspection. Reaches were excluded from analysis if the peak velocity during the reach fell more than 10% outside the bounds of the imposed velocity range (that is, if they were too fast or too slow). On average, this led to about 2.3% of all single-target reaches and 8.3% of all dual-target reaches being removed in Experiment 1, and 3.8% of single-target reaches and 8.6% of dual-target reaches removed in Experiment 2. In both cases, the majority of exclusions occurred because the latency was too long ($>400$ ms); for single-target trials, the remaining reaches were excluded mainly because they did not satisfy the block-specified speed range, while for dual-target trials the remaining excluded reaches were evenly divided between not satisfying the speed range and slowing down too early after triggering the target appearance. This suggests that in general, dual-target trials were more challenging for participants to perform because of a tendency to slow down after the correct target was cued, likely in an effort to make an appropriate corrective movement to hit the target. Critically however, there was no statistically significant difference in the number of reaches removed at each speed (two-sided paired t-test, Experiment 1, $p = 0.62$; Experiment 2, $p = 0.32$).

Velocity was computed by smoothing hand position using a second-order Savitzky-Golay filter with a frame size of 19 samples, then taking the numerical derivative of the smoothed position curves. Movement time was estimated to be the duration of time required for the participant to move 20 cm away from the starting position. Reaction time was computed as the time between the go cue and movement initiation (reported values were corrected for inherent delays in our equipment that have been measured to be $\sim 105$ ms). The initial reach direction was determined as the direction of the velocity vector 100 ms after movement initiation. The initial reach-direction error was computed as the angular difference between the initial reach direction and the direction of a straight line drawn between the start position and the correct target. The adjusted aiming direction was computed by first normalizing the absolute value of the initial reach error by the target-separation angle (such that a value of 0.5 is a reach directly aimed to either target and a value of zero is a reach directly between the two targets). A correlation was then assessed between reaction time and the adjusted aiming direction across all participants to determine if reaction times were longer for intermediate movements compared to reaches aimed directly at either of the potential targets.

Initial reach errors in dual-target trials were examined in two ways. First, we examined distributions of initial reach errors. For each speed and target separation, initial reach errors were pooled from all participants, then a Kolmogorov-Smirnov test was applied to compare differences in distributions between Fast and Slow movements for each target separation; $p$ values were corrected for multiple comparisons using a step-down Bonferroni-Holm method.

Second, to better characterize participants' behaviour, we sought to estimate the proportion of reaches directed either towards one of the two targets or intermediate between the targets. To perform this computation, we fit a mixture model

containing a unimodal Gaussian distribution (reflecting intermediate movements) and a bimodal Gaussian distribution (reflecting movements aimed directly to either target). We estimated the parameters of this model by maximum likelihood to obtain unbiased estimates of the relative proportion of intermediate movements. In fitting this model, there is a strong assumption that the underlying distribution is symmetric about the midline. However, participants may have generated asymmetric initial reach direction distributions, for example, if they opted to always reach in the same direction on every trial. To ensure a symmetric distribution for model fits, we used the initial reach error instead of the initial reach direction because the random choice of the correct target location on each trial regardless of the participant's behaviour ensured that the resulting initial-reach-error distribution was symmetric. Several parameters of this mixture model were constrained: we required that the two modes of the bimodal distribution (representing direct movements) be equidistant from the midline and have the same variance, consistent with the fact that the initial reach error was constructed to ensure a symmetric distribution. We also fixed the unimodal distribution (representing intermediate movements) to have its mean at the midline. Thus, the model contained four free parameters: the mean ($\mu_d$) and variance ($\sigma_d^2$) of the initial reach-direction error for direct reaches, the variance ($\sigma_i^2$) of the initial reach-direction error for intermediate reaches, and the overall proportion of movements that were intermediate ($\theta$). The probability density of observed reach errors was

$$P(\text{Initial Reach Error}) = (1 - \theta) \times p(\text{direct}) + \theta \times p(\text{intermediate})$$

$$= (1 - \theta) \times \left[ \frac{0.5}{\sqrt{2\pi\sigma_d^2}} e^{-\frac{(x - \mu_d)^2}{2\sigma_d^2}} + \frac{0.5}{\sqrt{2\pi\sigma_d^2}} e^{-\frac{(x + \mu_d)^2}{2\sigma_d^2}} \right] + \theta \times \left[ \frac{1}{\sqrt{2\pi\sigma_i^2}} e^{-\frac{x^2}{2\sigma_i^2}} \right].$$

The parameter of primary interest in this model was $\theta$, the proportion of intermediate movements generated (see Fig. 3a).

We used this model-based approach to estimate the proportion of intermediate movements produced by each individual participant using a three-stage fitting procedure to minimize the number of spurious model fits. In the first stage, the mixture model was fit to the pooled group data for each movement speed and target-separation angle to obtain stage-one estimates of distribution means and variances (that is, $\mu_d$, $\sigma_d^2$ and $\sigma_i^2$). These stage-one parameters are reported in Supplementary Table 1. In the second stage, the model was fit to each individual participant (that is, the distribution of reaches generated by a given participant for each movement speed and target-separation angle) to estimate $\theta$ while holding the other three parameters constant. Finally, to obtain estimates of the bias, the model was fit again for each participant while allowing only $\mu_d$ to vary (that is, the variances remained fixed and the value of $\theta$ was taken from the individual-level fits in the second stage). Code to fit these three-stage mixture models is available from the corresponding author on request.

Model parameters (that is, $\theta$) for each combination of speed and target-separation angle were compared across participants via mixed-effects models using the lme4 package in R[49], with p-values obtained using likelihood ratio tests of the full model including the effect of interest against the model without that effect. A post-hoc analysis examining the effect of target separation on $\theta$ was performed using a bootstrap approach whereby the model was refit to data for which the target separation angle labels were shuffled. In this way, a p-value was estimated for observing a main effect of a given magnitude. For Experiment 2, data were compared using two-sided paired t-tests as there was only one target-separation angle to consider.

As there is some concern that intermediate movements at the narrowest (15°) target-separation angle are difficult to distinguish from inherent reach variability, we repeated all analyses while excluding data from the 15° target-separation angle condition. In all cases exclusion of this data subset did not qualitatively change the results; thus, we report statistics including this condition throughout the manuscript.

We also examined how the likelihood of an intermediate movement ($\theta$) depended on differences in movement speed within each condition. For each instructed-speed condition (Fast or Slow), and for each target-separation angle, initial reach errors were pooled across all participants and sorted according to the hand speed at 4 cm (that is, the time that the correct target was revealed). Then, the mixture model was fit separately to all reaches that fell within a sliding window (width, 0.1 m s$^{-1}$) to estimate the proportion of intermediate movements, $\theta$, at each given reach speed (using group-level, stage-one values for the other three model parameters). To avoid overfitting to noisy, sparse data, the model was fit only if at least 50 data points fell within a given window. We formally tested whether speed affected behaviour within each condition by comparing estimates of the likelihood of an intermediate movement ($\theta$) within the slowest 25% and the fastest 25% of all reaches. We performed this comparison by estimating bootstrapped confidence intervals generated by resampling the data with replacement[50], and identified times when those confidence intervals were non-overlapping.

To estimate the likelihood of task success across different reaching speeds, we pooled the data across all target-separation angles and across both instructed speeds. We then identified, across all measured hand speeds at 4 cm, the range of movement speeds for which at least 20 trials were classified as intermediate and 20 trials were classified as direct according to the stage-two mixture model (that is, immediately after estimating $\theta$ for each subject). For this analysis, classification was assigned according to whether the likelihood of the reach being intermediate

(unimodal distribution) or direct (bimodal distribution) was greatest. Across a range of speeds, we examined the probability that reaches were successful given that they were classified as either intermediate or direct, and also calculated the proportion of trials that were assigned a classification of being intermediate.

Finally, we tested whether movement speed also affected reach direction in single-target trials. Although we did not expect behaviour to strongly resemble that in dual-target trials, it was possible that single-target reaches were biased toward the midline, and that this bias was speed-dependent. To quantify this effect, we calculated the bias in initial reach direction toward the midline for each participant as in Verstynen and Sabes[17]:

$$\text{Bias} = \frac{1}{N_{\text{Left}} + N_{\text{Right}}} \left( \sum_{i=1}^{N_{\text{Left}}} (\text{Initial Reach Error})_{\text{Left},i} - \sum_{i=1}^{N_{\text{Right}}} (\text{Initial Reach Error})_{\text{Right},i} \right).$$

Biases for each speed were compared across target-separation angles using a one-way ANOVA, and across speeds and target-separation angles using a two-way ANOVA. Additionally, for each participant single-target biases were compared to corresponding estimates of inward-directed biases of reaches aimed directly to the target on dual-target trials (based on the $\mu_d$ parameter from the mixture model) with a mixed-effects model using the lme4 package in R[49], with a p-value obtained using a likelihood ratio test.

**Data availability.** The data sets generated and analysed during the current study are available from the corresponding author on request.

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

## Acknowledgements

This work was supported by NSF grant BCS-1358756. We thank John Krakauer for helpful discussions.

## Author contributions

A.L.W. and A.M.H. designed the experiments. A.L.W. performed the research and analysed the data. A.L.W. and A.M.H. interpreted the results and wrote the paper.

## Additional information

**Competing financial interests:** The authors declare no competing financial interests.

