## [Peer Review File · Nature Communications]

Reviewers' Comments:

Reviewer #1 (Remarks to the Author):

Review of "Motor planning flexibly optimizes performance under uncertainty about task goals"

Reviewer: Mike Landy

This is a nice a pretty straightforward paper suggesting that, at least in their circumstance, a single motor plan is produced rather than multiple plans in parallel, when the eventual goal isn't known at movement onset. It's hard to complain about this conclusion, since the authors cite a paper of mine that comes to the same conclusion! ;^)

My comments are mostly minor.

Specifics (page/para/line):

5/2/11: I assume this is a correlation between RT and the absolute value of aiming direction, because otherwise this wouldn't make sense. But, it should say so explicitly here.

I wondered whether this strategy switch (go central, go toward the target) was a conscious, cognitive thing (seems likely), but I found it interesting in this regard that the use of the strategy varied with speed even in the slow condition. In our paper on this (cited here), the switch is continuous depending on the priors; here it's treated somewhat like a binary switch even though that seems doubtful. In Figure 4A (60 deg separation, for example) this looks like a very clear change of strategy with no intermediate plans/aims.

7/2/7: Subjects tended to use the strategy that would be more successful, but didn't use it 100% of the time. That could be noise in the decision about which strategy to use. Or it could be like probability matching (which is a bit of a myth, but more of a tendency to hedge one's bets on strategy).

8/2/3-4: This change in "bias" could well be a function of the 3-mode fit including more of the intermediate trials in the peripheral component of the fit, thus dragging them centrally. I don't think there's a good way to dismiss this objection.

10/1: This is pretty unclear. I didn't go back to reread Verstynen & Sabes, but in what sense is

this a prior? If it's a prior on target location (since we are talking about single-target trials), it should be pretty ineffective since visual uncertainty is tiny compared to the wide positioning of potential targets. And, this is a single-target trial, so there is no real uncertainty at the beginning of the reach as in two-target trials (for which a prior makes sense, as in our paper).

10/2: "Lingering uncertainty" is vague and seems more like a way to give the effect a name than having any explanatory power. Hmmmm.

13/2: I've read that Gallivan/Wolpert/... paper and agree that their data don't convince me of the two-plan hypothesis.

17/2/2: I was confused by this. 40% than what velocity? It seemed like the sample trials in Fig. 1C violated this, but maybe I misunderstood the baseline from which 40% was subtracted.

18/2/6: The fact that 16% of dual-target trials were removed compared to 10% of single-target seems like a result rather than something to hide in the Methods.

18/last: "first normalizing the IRTE by the target-separation angle": It's not completely clear what this means, so please spell it out or put an equation in which the denominator is clarified.

19/2: Is there any reason why you didn't fit 3 Gaussians (other than minimizing the number of parameters and/or making the fit procedure more stable)?

Figure 5: This figure is a bit odd. At first I misread the meaning of the figure, thinking that the changes of slope implied greater inclusion of the intermediate trials (the intermediate mode of the fit) into the peripheral modes (causing greater central bias). I think that's false, but some clarification would help. What's plotted is μ_b , and that should be stated on plot and/or legend. In B, clarify that negative means "outside".

Reviewer #2 (Remarks to the Author):

I think that this is overall a fantastic paper that shows that motor planning under uncertainty systematically depends on the speed of the planned movement. This effect is clear, and the experiments that demonstrate it are superbly designed. Moreover, as the authors maintain, the observed effect (that intermediate movements under target uncertainty systematically disappear as movement speed increases) has important implications for motor planning. However, I do think that the paper should be reframed as providing evidence against the motor averaging hypothesis rather than against the parallel planning hypothesis, as is currently the case. The conclusions which claim to refute the parallel planning hypothesis in favor of single planning

under uncertainty overreach the results, in my opinion. I do think that even after pulling back and refocusing the main claim, this paper presents a substantial advance in our understanding of motor planning, as evidence against the long-standing motor averaging hypothesis is quite important.

Specific:

The parallel planning hypothesis:

As mentioned above, I think that the argument the authors make supporting their claim that "Our findings refute the hypothesis of parallel planning" is flawed. Details below.

In the first two sentences of the abstract (copied below), I think the author mischaracterize the main evidence that the motor system simultaneously prepares for multiple potential goals.

"Existing theories posit that the motor system prepares for all potential goals by generating several movement plans in parallel. Such theories are founded on the observation that presenting two competing goals often results in a movement intermediate between these goals."

But the main evidence for the theory that that the motor system prepares for multiple potential goals is from neurophysiologic results rather than "the observation that presenting two competing goals often results in a movement intermediate between these goals", which comes from the motor averaging hypothesis. Thus the abstract makes it found like the current results directly challenge the theory that that the motor system prepares for multiple potential goals, whereas they do not.

Thus I think that the authors conclusion that "Our findings refute the hypothesis of parallel planning" isn't directly supported by this study. I think that the authors thus are overreaching here a bit, and that they should stick to assessing the motor averaging hypothesis as the parallel planning hypothesis is beyond the scope of the current study.

In particular, why is it not a logical possibility that individual plans could be generated for two (or, more generally, multiple) individual targets, but if there was still some uncertainty at movement onset, the movement be replanned to in light of this uncertainty? Or similarly, why is it not a possibility that individual plans could be generated for two single targets and for the uncertain movement onset case, if the motor system "knows" that target uncertainty might persist after movement onset because it has experienced this possibility before (as would be the case)? The framing and the discussion entirely fail to address these real and somewhat obvious possibilities. And, unless I'm mistaken, cannot distinguish between them and the single plan hypothesis favored by the authors.

Moreover, i do wonder more abstractly whether generating an important and testable distinction between planning in parallel for two potential goals (the parallel planning hypothesis) and generating a "single plan" under uncertainty where, for example, the uncertainty is characterized by two delta functions corresponding to the two potential goals.

The presentation of the data can be improved considerably:

In Figure 1 (particularly in 1d), the movement paths should be colored by the direction of the final target.

How was the "representative" participant chosen? Is she truly representative. I would be much better to show data from 2 or 3 randomly selected individuals (with randomly selected movements for each participant if the plot would be too dense to show all movements). This would allow the reader to have more confidence that the "representative" data was truly representative.

In Figure 2, i think that the highly smoothed pdf estimates without error bars that are shown stand in the way of letting the reader fully appreciate the data. Moreover the y-axis values of these smoothed pdf plots have the weird units of likelihood per degree (which have little meaning). In sum, I'm not a fan of this display. The authors should instead plot "regular" histograms with error bars. Error bars across individuals are crucial in assessing these data and the y-axis units would be more accessible (like # of movements, or fraction of the total movements in each bin). Additionally, the "smoothing" could be more transparently accomplished with the choice of bin width. Also, it is unnecessarily confusing that the x-axes of the two plots are different, greatly obscuring a direct comparison of the dual target trials in the top row and the single target trials in the bottom row. Right now the authors are comparing apples and oranges for the single target vs dual target trials in the figure - this should be fixed. The x-axis of the top row should be changed to "initial reach direction" to match the bottom row. Additionally, it would be good to indicate the left and right target directions in these plots. If the authors think it is important to plot reach error (and i think they probably should, presumably to demonstrate that participants truly don't "know" where the final target location will be) then they should do so in a separate panel for BOTH the single target and dual target trials to allow a direct comparison between the two.

Figures 3 & 4 should be combined. The two legends in 3a should be changed from unimodal vs bimodal to direct movements vs intermediate movements to make these plots more readily understandable. The current labels are about the model implementation rather than about what the model is trying to represent! The mixture model is probably fine i think, but i'm worried about one thing. If σ_u (the name of which should be changed to σ_i in one with the

comment about the plot legend above) were large enough then some movements that would fall under that distribution would in fact be aimed directly at one of the targets. This is odd, and not necessarily a problematic behavior, but it is at least one that could be argued about. A similar issue would occur if u_b were very small. But does any of this ever occur in the current data set? The problem is that right now the reader has absolutely no idea! The authors need to disclose this rather than withhold the model fit parameters from the reader, as is currently the case.

I'm not sure that the data in Figure 5 is worth being in a figure in the main paper. It's pretty tangential to the hypothesis, and is thus a bit distracting, in my opinion. I'd suggest putting this info in the main text only or putting it in the main text with a supplementary figure. But I'm not entirely opposed to keeping the figure if the authors maintain its importance.

Fairness in the 2nd section of the discussion

The 2nd section of the discussion "Biases on single-target trials can also be explained by the single-plan theory" is highly problematic in my view as it fails to meaningfully address how these biases might or might not be explained by the motor averaging hypothesis. It seems to me that the main idea supporting how the single-plan theory can explain single-target trial biases: "A priori uncertainty as to where the target will appear might still partially persist even after the target has been presented" could equally well be used to support how motor averaging can explain single-target trial biases. But this is never addressed. Thus I don't think that the single-target trial biases that the authors demonstrate meaningfully distinguish between the hypotheses at issue. This should be made clear. Or if the authors think otherwise, they should make a clear argument about this, rather than the one-sided nature of the current discussion on this point.

Minor:

The authors repeatedly use the word "interference" in very general vague manner that is highly confusing and possibly misleading. They should fix this. For instance, in the introduction they say: "This dependence on angular separation can, however, also be explained in terms of interference between competing plans: narrowly separated targets likely require movement plans with similar neural representations, and thus will be more prone to interference" But wouldn't movement plans that are more different "interfere" more (largely cancelling on another) whereas movement plans to targets with little angular separation would be extremely similar and thus could not "interfere" very much. I think what the authors mean here is something like:

"narrowly separated targets with movement plans having similar neural representations would interfere LESS, resulting in LESS cancellation when averaged together and thus predict more

pronounced intermediate movements by the motor averaging hypothesis"

The authors should either clearly define, up front, what they mean by interference and then use this term in a manner that is consistent, or change their word choice entirely.

it would be good to remove the IRE abbreviation. It only saves 22 words, at the expense of introducing unnecessary jargon.

The 3rd sentence of the discussion is unnecessarily confusing and should be revised.

Reviewer #3 (Remarks to the Author):

The authors test the hypothesis that the spatial averaging observed in go before you know paradigms is not attributable to multiple movement plans simultaneously co-existing, but to a single motor plan that seeks to optimize performance under goal uncertainty. They manipulate movement velocity and show that slow movements show spatial averaging much more so than fast movements. They argue that this supports the normative view that the motor system forms a single, flexible plan, optimized for uncertain goals.

The authors address a question (parallel planning) that has received a lot of interest in recent years. The work is carried out rigorously and the results are of potential interest. However, an aspect of the methodology is debatable, leading the authors to draw interpretations that are unsupported by the data. In this light they cannot really speak to parallel planning.

A fundamental difference between this paper and previous ones i.e. Chapman et al. (2010) is the fact that targets were presented a long time before the go-signal (here 400-1000ms instead of RTs within 325 ms in Chapman). Spatial averaging is particularly convincing in contexts where subjects are forced to commit rapidly after target presentation, leaving no time to deliberate strategically as to where to initiate the reach. In the present context there was plenty of time for strategy to play in, possibly overriding any parallel encoding. It is possible that parallel planning is a transient phenomenon, being amenable to top-down modulations when more time is available for deliberation, allowing the brain to choose a specific (optimal) course of action. The more important point here is that while these results may speak to subjects using different strategies as a function of movement speed (which is not uninteresting in itself), they don't invalidate the possibility that the brain could simultaneously represent multiple motor plans.

That being said it would be interesting to redo this experiment while forcing subjects to commit rapidly. One hint that different results may arise is the fact that Chapman et al. (2010) also used very rapid MTs (under 425ms) and yet subjects demonstrated strong spatial averaging. How does that compare to the straight trajectories here in the fast condition?

The hypothesis put forth in the introduction (i.e. if intermediate movements reflect a purposeful strategy, participants would generate them more often when moving slowly compared to moving quickly - page 5) is not exclusive and is debatable. In fact one could argue that when given more time to correct, subjects could commit to one target (hence taking a 50% chance that they would not have to correct if they happened to choose the right one), and then still have ample time to perform a large online correction if needed.

If, as the authors put forth, the motor system generates a single motor plan, then how do they reconcile the neurophysiological evidence showing activity tuned to multiple targets in PMd and PPC? In this regards, the proposed nuance between motor goal and motor plan should be more convincingly backed up from a neural instantiation standpoint, otherwise it comes through as just semantics. The debate between sensory and motor has been addressed (Gallivan papers) - so here motor goal must not be simply sensory?

Please provide MT directly in the main paper. I can get from fig 1c that it is about 0.5 and 1s. This paper critically hinges on the time available to correct the movement.

Minor

Many references don't contain the year.

Response to Reviewers' Comments

Reviewers' comments:

Reviewer #1 (Remarks to the Author):

Review of "Motor planning flexibly optimizes performance under uncertainty about task goals"

Reviewer: Mike Landy

This is a nice a pretty straightforward paper suggesting that, at least in their circumstance, a single motor plan is produced rather than multiple plans in parallel, when the eventual goal isn't known at movement onset. It's hard to complain about this conclusion, since the authors cite a paper of mine that comes to the same conclusion! ;^)

My comments are mostly minor.

Specifics (page/para/line):

5/2/11: I assume this is a correlation between RT and the absolute value of aiming direction, because otherwise this wouldn't make sense. But, it should say so explicitly here.

Correct. Note also that this correlation is not with the actual aiming direction, but with a normalized value that accounts for both the target-separation angle and treats distance away from the midline as the independent variable. This was mentioned briefly in the methods section, but we now describe it more clearly and also briefly note it in the results section.

I wondered whether this strategy switch (go central, go toward the target) was a conscious, cognitive thing (seems likely), but I found it interesting in this regard that the use of the strategy varied with speed even in the slow condition. In our paper on this (cited here), the switch is continuous depending on the priors; here it's treated somewhat like a binary switch even though that seems doubtful. In Figure 4A (60 deg separation, for example) this looks like a very clear change of strategy with no intermediate plans/aims.

We agree that this strategy switch may be very cognitive; this could explain why the switch in behaviors appears so binary. In fact, the results from Experiment 2 suggest that this is likely to be true, since reducing the time available to deliberate about the presence of two potential stimuli tends to reduce the proportion of intermediate movements generated.

The binary nature of the direct/intermediate behavior is not really surprising as there is no good reason to generate a reach that is aimed in a direction other than directly to the target or directly in between the two targets; because the target choice was completely random there is no obvious reason to favor one direction or the other (i.e., introduce a bias) - unlike in your paper and others where asymmetric target odds promote a directional bias.

7/2/7: Participants tended to use the strategy that would be more successful, but didn't use it 100% of the time. That could be noise in the decision about which strategy to use. Or it could be

like probability matching (which is a bit of a myth, but more of a tendency to hedge one's bets on strategy).

We agree that it is unclear why participants did not use the more successful strategy 100% of the time, or in fact why some subjects never attempt the more successful strategy at all. Another potential explanation is that there may simply be unintended variability in the speed at which the movement is executed, after a strategy has been selected. We now acknowledge briefly in the discussion that we do not know exactly what guides the choice of a particular response on any given trial or by any given subject.

8/2/3-4: This change in "bias" could well be a function of the 3-mode fit including more of the intermediate trials in the peripheral component of the fit, thus dragging them centrally. I don't think there's a good way to dismiss this objection.

At small target separations this is indeed an issue, particularly for the 15° separation where the components of the model are very close to one another. However, at very large separations (45°, 60°), this problem no longer occurs. The fact that there is a bias is quite evident in the group histograms (Figure 2), so we do not believe this result could possibly be artifactual.

Nevertheless, we did carefully re-examine the model fits for all subjects in light of this concern (and similar concerns from Reviewer 2) and found that there were a small minority of subjects for whom the model was overfitted (1-3 per group). This was primarily for subjects/conditions which exhibited no direct movements at all – in which case the 'direct' movement component was more prone to getting biased more centrally. This affected both our estimate of θ (the proportion of intermediate movements) and our estimate of the bias. In order to obtain more robust estimates for these subjects, we altered our fitting procedure as follows: First, we fit the full mixture model to the group data. Next, we obtained estimates of the number of intermediate movements for each subject by allowing θ to vary for each subject, but fixing all other parameters to be equal to the group fit. To estimate the bias, we eliminated any subjects who did not generate enough direct movements ($\theta < 0.1$), and then re-fitted the model for the remaining subjects, allowing only the direct-movement bias μ_d to vary.

This procedure was effective in avoiding the overfitting problem. The overall quantitative change to the results is quite minor, but we believe this improved analysis method provides a more precise reflection of participants' behavior. Our core statistical results and conclusions were not affected by this revision to the analysis.

10/1: This is pretty unclear. I didn't go back to reread Verstynen & Sabes, but in what sense is this a prior? If it's a prior on target location (since we are talking about single-target trials), it should be pretty ineffective since visual uncertainty is tiny compared to the wide positioning of potential targets. And, this is a single-target trial, so there is no real uncertainty at the beginning of the reach as in two-target trials (for which a prior makes sense, as in our paper).

Verstynen and Sabes do suggest that participants maintain a prior over target locations (indeed 'prior' is in the title of their paper). They report that the pattern of biases in reach direction were consistent with a Bayesian estimate of the target location, with a likelihood with $\sigma \approx 7^\circ$, which

does not seem implausible if one considers that this may also involve uncertain coordinate transformations. Our results for the Slow condition are consistent with this Bayesian theory, however, the fact that these biases are eliminated when moving quickly refutes this theory.

10/2: "Lingering uncertainty" is vague and seems more like a way to give the effect a name than having any explanatory power. Hmmmm.

Agreed. We have revised this section to clarify our thoughts and have removed this vague phrasing. We now simply say that prior uncertainty might still exert an influence on behavior even after the target has been seen.

13/2: I've read that Gallivan/Wolpert/... paper and agree that their data don't convince me of the two-plan hypothesis.

We appreciate that the reviewer agrees with our assessment.

17/2/2: I was confused by this. 40% than what velocity? It seemed like the sample trials in Fig. 1C violated this, but maybe I misunderstood the baseline from which 40% was subtracted.

We apologize for the confusion. The point of this criterion was that we did not want participants to make an initial short, rapid movement burst to trigger the target appearance (particularly in the "Fast" condition) and then slow down to give themselves time to generate movement corrections. Thus, we imposed a requirement that the reach velocity could never decrease to less than 40% of the maximum velocity exhibited thus far during the trial.

18/2/6: The fact that 16% of dual-target trials were removed compared to 10% of single-target seems like a result rather than something to hide in the Methods.

We apologize for an arithmetic error that we identified underlying these reported values. The correct values are that, for dual-target trials, on average 8.3% of all reaches were removed; for single-target trials, 2.3% of all reaches were removed (mean, ~5% across all conditions). Dual-target trials tended to be rejected because subjects moved too slowly – perhaps a natural tendency when moving under uncertainty and attempting to make successful movement corrections. Indeed, the highest success rates we observed were for intermediate movements generated at slow movement speeds. We now describe this in the Methods.

18/last: "first normalizing the IRTE by the target-separation angle": It's not completely clear what this means, so please spell it out or put an equation in which the denominator is clarified.

We have added some text to clarify exactly how this normalization was performed.

19/2: Is there any reason why you didn't fit 3 Gaussians (other than minimizing the number of parameters and/or making the fit procedure more stable)?

We did want to minimize the number of free parameters to avoid overfitting, particularly given that we were fitting the model separately for individual participants and conditions, with limited

data. The distributions of the initial reach errors for the direct components should be symmetric. Subjects were guessing in these trials, thus the magnitude of the components should be the same on each side (indeed that is why we analyze initial reach error rather than initial reach direction). These also include reaches from both directions, eliminating any asymmetry that might arise from biomechanical differences. Therefore, enforcing symmetry on the direct components of the model enabled us to fit a simpler model without incurring any meaningful loss of generality. We now clarify this reasoning in our methods.

Figure 5: This figure is a bit odd. At first I misread the meaning of the figure, thinking that the changes of slope implied greater inclusion of the intermediate trials (the intermediate mode of the fit) into the peripheral modes (causing greater central bias). I think that's false, but some clarification would help. What's plotted is μ_b , and that should be stated on plot and/or legend. In B, clarify that negative means "outside".

We apologize for the confusion. This figure actually compares two distinct analyses; estimates of bias on single-target trials are calculated according to the Verstynen and Sabes bias equation noted in the methods, whereas on dual-target trials the bias comes from the estimated μ_b (now μ_d) parameter of the mixture model. We now clarify where the dual-target bias comes from in the figure legend. We also note what positive and negative mean.

As for the interpretation, the reviewer is correct. The y axis does not reflect anything about intermediate movements (indeed there are no intermediate movements in the single-target case), but instead reflects the tendency for direct movements to be slightly biased towards the midline. We have modified the figure and the text in the results section to help clarify this.

Reviewer #2 (Remarks to the Author):

I think that this is overall a fantastic paper that shows that motor planning under uncertainty systematically depends on the speed of the planned movement. This effect is clear, and the experiments that demonstrate it are superbly designed. Moreover, as the authors maintain, the observed effect (that intermediate movements under target uncertainty systematically disappear as movement speed increases) has important implications for motor planning. However, I do think that the paper should be reframed as providing evidence against the motor averaging hypothesis rather than against the parallel planning hypothesis, as is currently the case. The conclusions which claim to refute the parallel planning hypothesis in favor of single planning under uncertainty overreach the results, in my opinion. I do think that even after pulling back and refocusing the main claim, this paper presents a substantial advance in our understanding of motor planning, as evidence against the long-standing motor averaging hypothesis is quite important.

Thank you for the positive comments. We have followed the suggestion to refocus the main claims of the manuscript on movement averaging rather than parallel planning. We agree that our findings speak most directly to motor averaging. However, we maintain a substantial discussion of parallel planning, because these are closely related themes and we believe our findings are pertinent to this theory.

We also now include a fuller discussion of the neural evidence in favor of parallel planning – which we believe is quite weak because i) parallel activity construed as plans could just as easily represent goals and ii) brain regions that do clearly represent movement plans rather than goals appear to only support representation of a single plan at a time (e.g., Churchland et al).

Specific:

The parallel planning hypothesis:

As mentioned above, I think that the argument the authors make supporting their claim that "Our findings refute the hypothesis of parallel planning" is flawed. Details below.

In the first two sentences of the abstract (copied below), I think the author mischaracterize the main evidence that the motor system simultaneously prepares for multiple potential goals.

"Existing theories posit that the motor system prepares for all potential goals by generating several movement plans in parallel. Such theories are founded on the observation that presenting two competing goals often results in a movement intermediate between these goals."

But the main evidence for the theory that that the motor system prepares for multiple potential goals is from neurophysiologic results rather than "the observation that presenting two competing goals often results in a movement intermediate between these goals", which comes from the motor averaging hypothesis. Thus the abstract makes it found like the current results directly challenge the theory that that the motor system prepares for multiple potential goals, whereas they do not.

Thus I think that the authors conclusion that "Our findings refute the hypothesis of parallel planning" isn't directly supported by this study. I think that the authors thus are overreaching here a bit, and that they should stick to assessing the motor averaging hypothesis as the parallel planning hypothesis is beyond the scope of the current study.

It is true that neurophysiological findings first prompted the parallel planning hypothesis. However, these findings are far from definitive evidence of parallel planning. As we explain in the discussion, parallel encoding could well represent movement goals (i.e. a cost function), rather than a plan for the movement itself. As we also now note in the discussion, current theories of movement planning in the motor cortex (primarily the work of Churchland and Shenoy) suggest that the motor cortex encodes just a single movement policy at a time – incompatible with the parallel planning hypothesis.

Perhaps in testimony to the weakness of the neurophysiological evidence, there have been a large number of recent publications on this topic, all demonstrating variations on movement averaging and asserting that this is evidence for the existence of parallel motor plans. Refuting movement averaging effectively disqualifies this behavioral line of evidence for parallel planning, leaving only neurophysiological evidence, which is far from clear.

In particular, why is it not a logical possibility that individual plans could be generated for two (or, more generally, multiple) individual targets, but if there was still some uncertainty at movement onset, the movement be replanned in light of this uncertainty? Or similarly, why is it not a possibility that individual plans could be generated for two single targets and for the uncertain movement onset case, if the motor system "knows" that target uncertainty might persist after movement onset because it has experienced this possibility before (as would be the case)? The framing and the discussion entirely fail to address these real and somewhat obvious possibilities. And, unless I'm mistaken, cannot distinguish between them and the single plan hypothesis favored by the authors.

Moreover, I do wonder more abstractly whether generating an important and testable distinction between planning in parallel for two potential goals (the parallel planning hypothesis) and generating a "single plan" under uncertainty where, for example, the uncertainty is characterized by two delta functions corresponding to the two potential goals.

We agree that, based on our behavioral data alone, we cannot rule out the possibility that multiple motor plans are prepared, of which all but one (the most efficient one that specifies an intermediate movement) is ultimately discarded. Indeed, it may be impossible to completely refute the notion of parallel planning through behavioral experiments – one can always speculate the existence of ‘covert’ plans that exert little or no influence over behavior.

We have therefore softened the claims in our paper as regards parallel planning and have removed claims that our data “refute” parallel planning. We nevertheless still include a substantial discussion of the parallel planning theory, pointing out the limitations of existing behavioral and neural evidence in favor of it.

The presentation of the data can be improved considerably:

In Figure 1 (particularly in 1d), the movement paths should be colored by the direction of the final target.

Revised. We have now modified Figure 1D to color-code the movement paths according to the direction of the final target; reaches to the right are darker in shade compared to reaches to the left.

How was the "representative" participant chosen? Is she truly representative. I would be much better to show data from 2 or 3 randomly selected individuals (with randomly selected movements for each participant if the post would be too dense to show all movements). This would allow the reader to have more confidence that the "representative" data was truly representative.

We agree that showing data from just a single subject is problematic. However, we are concerned that Figure 1 would become too crowded if it included data from multiple participants. Instead, we now provide data from all 16 participants (displayed as in Figure 1D) in Supplemental Figure 1, so readers may examine the complete data set. Also note that much of

the analyses and figures that follow (Figure 2, Figure 3B) clearly demonstrate that the group-level behavior is consistent with that of this one individual.

In Figure 2, I think that the highly smoothed pdf estimates without error bars that are shown stand in the way of letting the reader fully appreciate the data. Moreover the y-axis values of these smoothed pdf plots have the weird units of likelihood per degree (which have little meaning). In sum, I'm not a fan of this display. The authors should instead plot "regular" histograms with error bars. Error bars across individuals are crucial in assessing these data and the y-axis units would be more accessible (like # of movements, or fraction of the total movements in each bin). Additionally, the "smoothing" could be more transparently accomplished with the choice of bin width.

We have followed this suggestion. The pdf estimates originally presented did not have error bars because they reflected data pooled from all participants, not average data. We now present "regular" averaged histograms across subjects, with error bars.

Also, it is unnecessarily confusing that the x-axes of the two plots are different, greatly obscuring a direct comparison of the dual target trials in the top row and the single target trials in the bottom row. Right now the authors are comparing apples and oranges for the single target vs dual target trials in the figure - this should be fixed. The x-axis of the top row should be changed to "initial reach direction" to match the bottom row. Additionally, it would be good to indicate the left and right target directions in these plots. If the authors think it is important to plot reach error (and i think they probably should, presumably to demonstrate that participants truly don't "know" where the final target location will be) then they should do so in a separate panel for BOTH the single target and dual target trials to allow a direct comparison between the two.

In these figures, we in fact deliberately chose the x axes to be different in order to enable a fairer comparison between the distributions in the dual- and single-target conditions. In the dual-target condition, is it possible for individual participants to adopt an asymmetric strategy, always aiming directly for the same target on every trial. Plotting the initial reach error rendered this distribution symmetric, since the subject had a 50-50 chance of guessing the correct target direction. This therefore allowed a clearer visual comparison with the distribution of reach directions for single-target trials, which is inherently symmetric by design. Note that, as a result, we cannot strictly label the axes as left and right for the dual-target trials. To aid comparison, we then specified the limits of the x-axis such that the locations of the targets would line up in the two panels. We also now include gray lines indicating the locations of the targets to facilitate this comparison between dual-target and single-target trials.

Figures 3 & 4 should be combined.

We prefer to keep Figures 3 and 4 separate since they represent different analyses and make distinct points about the data.

The two legends in 3a should be changed from unimodal vs bimodal to direct movements vs intermediate movements to make these plots more readily understandable. The current labels are about the model implementation rather than about what the model is trying to represent!

We fully agree. Thank you for pointing this out. We have changed the axis labels and notation throughout, as well as the way we present the results in the text to reflect what the model represents rather than its implementation (e.g. “the estimated variability of direct movements” rather than “width of the the bimodal mixture component”).

The mixture model is probably fine i think, but i'm worried about one thing. If σ_u (the name of which should be changed to σ_i in one with the comment about the plot legend above) were large enough then some movements that would fall under that distribution would in fact be aimed directly at one of the targets. This is odd, and not necessarily a problematic behavior, but it is at least one that could be argued about. A similar issue would occur if u_b were very small. But does any of this ever occur in the current data set? The problem is that right now the reader has absolutely no idea! The authors need to disclose this rather than withhold the model fit parameters from the reader, as is currently the case.

We were also concerned about overlap between distributions. In practice, the components of the mixture model overlapped very little, apart from in the 15° separation case, which certainly compromises our ability to discern between direct and intermediate reaches. We went back and carefully examined the model fit for each subject, and noted a small subset of poor fits. To address this, we modified our model-fitting procedure using a three-stage fitting process. In the first stage, we fit the mixture model to the group-level data, which provided a more robust estimate of the mean and variance parameters. In the second stage, we used the values obtained from the group-level estimates to fit data from individual participants, while allowing only the proportion of intermediate movements θ to vary. This afforded more robust estimates of θ for each subject. Finally in the third stage, we estimated each subject's bias by re-fitting the model to each individual subject while allowing only μ_d to vary (i.e., the variances remained fixed using the group-level parameter estimates, and the θ value was taken from the individual-participant fits in stage 2). Whenever μ_d could not be well estimated due to a paucity of direct movements (i.e., when $\theta > 0.9$), we removed this participant's data from the bias estimate for that condition. This new fitting procedure did not qualitatively change any of the main findings presented previously in the manuscript; however, it did reveal a slight reduction in the proportion of intermediate movements as target-separation angle increased (when either including or excluding the 15° separation case).

For completeness, we now report group-level model parameters in Supplemental Table 1.

I'm not sure that the data in Figure 5 is worth being in a figure in the main paper. It's pretty tangential to the hypothesis, and is thus a bit distracting, in my opinion. I'd suggest putting this info in the main text only or putting it in the main text with a supplementary figure. But I'm not entirely opposed to keeping the figure if the authors maintain its importance.

Although this result is somewhat tangential, we believe this is an important result that will be of interest to many readers. We prefer to keep this figure in the main manuscript to ensure that the result is clearly visible. Moreover, these findings could potentially be explained using the same hypothesis of planning under uncertainty (rather than needing to invoke a separate idea about use-dependent plasticity), so we feel it is reasonably in-keeping with the main hypothesis.

Fairness in the 2nd section of the discussion

The 2nd section of the discussion "Biases on single-target trials can also be explained by the single-plan theory" is highly problematic in my view as it fails to meaningfully address how these biases might or might not be explained by the motor averaging hypothesis. It seems to me that the main idea supporting how the single-plan theory can explain single-target trial biases: "A priori uncertainty as to where the target will appear might still partially persist even after the target has been presented" could equally well be used to support how motor averaging can explain single-target trial biases. But this is never addressed. Thus I don't think that the single-target trial biases that the authors demonstrate meaningfully distinguish between the hypotheses at issue. This should be made clear. Or if the authors think otherwise, they should make a clear argument about this, rather than the one-sided nature of the current discussion on this point.

We agree that the single-target biases do not provide additional evidence towards the main hypothesis. We wanted to make the point that these biases share a strong qualitative similarity (speed-dependence) with intermediate movements and thus may derive from a common mechanism. We have modified the tone of this section to clarify this and point out that theories that account for intermediate movements when planning under uncertainty (whether through a single plan or through movement averaging) can also encompass use-dependent biasing effects.

Minor:

The authors repeatedly use the word "interference" in very general vague manner that is highly confusing and possibly misleading. They should fix this. For instance, in the introduction they say: "This dependence on angular separation can, however, also be explained in terms of interference between competing plans: narrowly separated targets likely require movement plans with similar neural representations, and thus will be more prone to interference" But wouldn't movement plans that are more different "interfere" more (largely cancelling on another) whereas movement plans to targets with little angular separation would be extremely similar and thus could not "interfere" very much. I think what the authors mean here is something like:

"narrowly separated targets with movement plans having similar neural representations would interfere LESS, resulting in LESS cancellation when averaged together and thus predict more pronounced intermediate movements by the motor averaging hypothesis"

The authors should either clearly define, up front, what they mean by interference and then use this term in a manner that is consistent, or change their word choice entirely.

We acknowledge that the choice of the word 'interference' here was not a good one because the term is so overloaded and we already used the term movement averaging to refer to essentially the same thing (albeit at a less mechanistic level of description). We have revised the text throughout to refer simply to "movement averaging" throughout.

When we referred to interference, we meant the process that leads to movement averaging, though in particular we had in mind the kind of low-level planning theories of Erlhagen and Schoner and of Cisek. In these theories, units (neurons) with similar tuning (planned movement

directions) have mutual excitation and hills of such activity are prone to coalesce (i.e., interfere), leading to an intermediate movement, while neurons with very different tuning mutually inhibit one another, leading to a winner-take-all dynamic (i.e. selection). This type of theory is popular because it can explain the effect of target separation on movement averaging.

It would be good to remove the IRE abbreviation. It only saves 22 words, at the expense of introducing unnecessary jargon.

We have made this suggested change.

The 3rd sentence of the discussion is unnecessarily confusing and should be revised.

The entire discussion has now been revised.

Reviewer #3 (Remarks to the Author):

The authors test the hypothesis that the spatial averaging observed in go before you know paradigms is not attributable to multiple movement plans simultaneously co-existing, but to a single motor plan that seeks to optimize performance under goal uncertainty. They manipulate movement velocity and show that slow movements show spatial averaging much more so than fast movements. They argue that this supports the normative view that the motor system forms a single, flexible plan, optimized for uncertain goals.

The authors address a question (parallel planning) that has received a lot of interest in recent years. The work is carried out rigorously and the results are of potential interest. However, an aspect of the methodology is debatable, leading the authors to draw interpretations that are unsupported by the data. In this light they cannot really speak to parallel planning.

A fundamental difference between this paper and previous ones i.e. Chapman et al. (2010) is the fact that targets were presented a long time before the go-signal (here 400-1000ms instead of RTs within 325 ms in Chapman). Spatial averaging is particularly convincing in contexts where participants are forced to commit rapidly after target presentation, leaving no time to deliberate strategically as to where to initiate the reach. In the present context there was plenty of time for strategy to play in, possibly overriding any parallel encoding. It is possible that parallel planning is a transient phenomenon, being amenable to top-down modulations when more time is available for deliberation, allowing the brain to choose a specific (optimal) course of action. The more important point here is that while these results may speak to participants using different strategies as a function of movement speed (which is not uninteresting in itself), they don't invalidate the possibility that the brain could simultaneously represent multiple motor plans.

That being said it would be interesting to redo this experiment while forcing participants to commit rapidly. One hint that different results may arise is the fact that Chapman et al. (2010) also used very rapid MTs (under 425ms) and yet participants demonstrated strong spatial averaging. How does that compare to the straight trajectories here in the fast condition?

Thank you for raising this very important concern. Our original experiment was designed to replicate most closely the paradigms used by Gallivan and Flanagan, who also allowed long preparation times (750 ms to 2 s) and drew strong conclusions about parallel planning based on observing intermediate movements. We have now performed a second experiment (Experiment 2 in the revised manuscript) that is more similar to the paradigm used by Chapman and colleagues: the go cue was synchronous with target presentation, and we enforced a reaction time limit of 400 ms. The findings from this new experiment were qualitatively similar to our original experiment. We did not observe any ‘unmasking’ of movement averaging for fast movements when reaction times were shortened. If anything, limiting reaction times tended to reduce the overall proportion of intermediate movements at both Fast and Slow movement speeds. Participants were thus capable of flexibly altering their behavior even when preparation time was limited, and we conclude that this flexibility is not purely the result of a deliberative process masking a more low-level outcome.

We note two important differences between the Chapman and Goodale paper and ours that might account for apparent discrepancies in the results:

1. Speed requirements: Unlike the Chapman and Goodale paper, we imposed strict velocity criteria on reaches such that subjects could not initiate their movement very slowly and then dramatically speed up during their movement once they were certain of the target location (especially in the “Fast” condition), nor could they begin very quickly and slow down to provide additional time to implement a movement correction. We used shooting movements that encourage participants to maintain their speed all the way through the target, unlike Chapman and Goodale who employed point-to-point movements that required subjects to decelerate and stop on the target. Thus, although the speed of reaching was likely comparable between their experiments and ours, the “Fast” movements required in our paradigm may have actually been more stringent and afforded less decision time compared to the Chapman and Goodale paradigm (particularly in our Experiment 2).

2. Target separation angle: Chapman and Goodale used targets separated by 12 cm, positioned on a screen 40 cm from the subject; this corresponds roughly to a target-separation angle of 17 degrees. In our data, we found that it was difficult to clearly distinguish between direct and intermediate reaches with such a close target separation (15 degrees, in our Experiment 1) because the distributions of reaches aimed directly to the target and intermediate to the targets highly overlapped. Nevertheless, we do know from ours and other studies that intermediate movements are more advantageous at narrow separations, and empirically appear more likely to be generated at smaller target-separation angles. This may explain why Chapman and Goodale saw such a high frequency of intermediate reaches in their experiment even with restricted planning times.

The hypothesis put forth in the introduction (i.e. if intermediate movements reflect a purposeful strategy, participants would generate them more often when moving slowly compared to moving quickly - page 5) is not exclusive and is debatable. In fact one could argue that when given more time to correct, participants could commit to one target (hence taking a 50% chance that they would not have to correct if they happened to choose the right one), and then still have ample

time to perform a large online correction if needed.

We believe the hypothesis is quite clear – participants will generate more intermediate movements when moving slowly than quickly. It is true that had we observed mostly direct movements at both fast and slow speeds, it would have been difficult to draw meaningful conclusions about movement averaging or parallel planning. However, we know from many prior studies that participants favor intermediate movements at regular speeds, and we and others have pointed out a clear normative rationale for this behavior (e.g., Haith et al. Plos Comp Biol 2015, Hudson et al. J Neurophysiol 2007). Our prediction was a logical extension of this rationale. Our findings reveal that even when moving at very Slow speeds, subjects favor generating intermediate movements instead of committing to one target and then correcting if they are wrong. Furthermore, the analysis of success rates of different movement strategies validates our core assumption that an intermediate movement strategy is more successful than a direct one at slow speeds, but less successful at fast speeds.

If, as the authors put forth, the motor system generates a single motor plan, then how do they reconcile the neurophysiological evidence showing activity tuned to multiple targets in PMd and PPC? In this regards, the proposed nuance between motor goal and motor plan should be more convincingly backed up from a neural instantiation standpoint, otherwise it comes through as just semantics. The debate between sensory and motor has been addressed (Gallivan papers) - so here motor goal must not be simply sensory?

We have tried to articulate the distinction between goals and plans more clearly in the revised manuscript. A goal defines the desired outcome of a movement without specifying how exactly to perform the action to achieve it. A motor plan, by contrast, details exactly how the action will be carried out to attain the goal. In terms of optimal control theory, the distinction between goals and plans is equivalent to that between cost functions and control policies.

A goal is indeed not simply a sensory representation of a target; for example, it is possible to have a motor goal of moving partway to a target (Bahcall and Kowler 2000) or to move to a position that is equal in magnitude but opposite in direction from the target (i.e., an antisaccade or anti-reach).

We have tried to more fully articulate how this distinction might be relevant to the interpretation of neural data. Specifically, we note that the simultaneous activity identified by Cisek and Kalaska (recorded from anterior PMd) could plausibly represent either goals or plans. It is, however, known (based on the work of Churchland and Shenoy) that more posterior regions of PMd (closer to M1 and the central sulcus) represent quite detailed information about an upcoming movement, i.e. they clearly do represent a movement plan. Interestingly, current theories of how plans are encoded in this region suggest that only a single motor plan can be encoded at any given time – contradicting the parallel planning hypothesis.

Please provide MT directly in the main paper. I can get from fig 1c that it is about 0.5 and 1s. This paper critically hinges on the time available to correct the movement.

Note that participants were not required to stop on the target but instead would often shoot through it. Thus, movement time could vary widely depending on the exact amplitude of the outward movement generated. The most meaningful measure of movement time is the time that it took the subject to move 20 cm away from the starting position, which is approximately 0.3 and 0.6 seconds respectively. However, note that Chapman and Goodale required their subjects to reach a farther distance, and moreover do not report peak movement speed; thus, we find it challenging to directly compare reaches between these two papers. Nevertheless, we now report movement time in our manuscript.

Minor

Many references don't contain the year.

We apologize for this error, the reference manager seems to have formatted some of the citations oddly, making it appear that some citations did not have years included. This has now been corrected.

Reviewers' Comments:

Reviewer #1 (Remarks to the Author):

Reviewer: Michael Landy

This version addresses my other minor comments, but having re-read the paper, I've got a couple more:

line 122: The word "influenced" implies causality (rather than, e.g., sharing a common other cause), and the evidence doesn't support that interpretation.

lines 180 and 458: The justification for Expt. 2 was to avoid subjects using cognitive strategy to, e.g., switch between direct and intermediate conditions. However, this experiment included only a single eccentricity condition, so that observers could (immediately, or within a few trials) decide consciously to perform an intermediate reach if two targets appear and a direct reach if only one does. Sure, you still get increase in intermediate reaches with speed within a block (e.g., within the SLOW block). But surely you can't suggest that speed from reach to reach is random, but having been randomly chosen, it THEN leads to a decision to reach directly or intermediate. So, the logic of how this works and how it isn't possibly cognitively controlled is unconvincing. I can easily imagine deciding before a trial to reach faster and use an intermediate reach if two targets appear.

Reviewer #3 (Remarks to the Author):

The authors have done an ok job with the revisions. The manuscript is enhanced thanks to experiment 2. However, it is surprising that the authors do not discuss their results in light of the recent Gallivan et al. (2016) Nat Neurosci paper which argues for the parallel specification of competing sensorimotor control policies for alternative action options. The fact that multiple control policies can be computed in parallel is particularly relevant to the motor goal/plan discussion on page 15.

line 254: must HAVE been

Figure 5: the dotted line seems to be the wrong color in the legend

Response to Reviewers' Comments

Reviewers' comments:

Reviewer #1 (Remarks to the Author)

Reviewer: Michael Landy

This version addresses my other minor comments, but having re-read the paper, I've got a couple more:

line 122: The word "influenced" implies causality (rather than, e.g., sharing a common other cause), and the evidence doesn't support that interpretation.

Agreed. We have changed this to "correlated with" to remain neutral with regard to causality.

lines 180 and 458: The justification for Expt. 2 was to avoid subjects using cognitive strategy to, e.g., switch between direct and intermediate conditions. However, this experiment included only a single eccentricity condition, so that observers could (immediately, or within a few trials) decide consciously to perform an intermediate reach if two targets appear and a direct reach if only one does. Sure, you still get increase in intermediate reaches with speed within a block (e.g., within the SLOW block). But surely you can't suggest that speed from reach to reach is random, but having been randomly chosen, it THEN leads to a decision to reach directly or intermediate. So, the logic of how this works and how it isn't possibly cognitively controlled is unconvincing. I can easily imagine deciding before a trial to reach faster and use an intermediate reach if two targets appear.

We apologize for the confusion with regard to the justification for this experiment, and have clarified this in the manuscript. This experiment was conducted to address the concerns of Reviewer 3, namely that motor averaging might have been transiently present for fast movements, but was masked by the fact that we allowed relatively long preparation times, allowing participants time to adopt a more cognitive strategy. The results clearly demonstrate that this was not the case: participants generated direct movements at fast speeds even at low RTs.

We agree that the most parsimonious explanation is that participants were able to flexibly adopt whichever movement strategy they preferred. In other words, they were not bound by any low-level, involuntary movement-averaging effects.

Reviewer #3 (Remarks to the Author)

The authors have done an ok job with the revisions. The manuscript is enhanced thanks to experiment 2. However, it is surprising that the authors do not discuss their results in light of the recent Gallivan et al. (2016) Nat Neurosci paper which argues for the parallel specification of competing sensorimotor control policies for alternative action options. The fact that multiple

control policies can be computed in parallel is particularly relevant to the motor goal/plan discussion on page 15.

We had initially removed discussion of the Gallivan et al. 2016 Nat Neurosci particular paper because we did not want to labor the discussion section by describing details of too much other work. However, we agree with the reviewer that a brief discussion of this paper is important and have added it back to our manuscript. In general, we feel that Gallivan et al. 2016 is not a convincing demonstration of parallel specification of competing control policies because it, along with other examples previously mentioned in our discussion, cannot clearly distinguish between the motor averaging and the normative hypotheses. The normative hypothesis would predict that because high feedback gains are more effortful, participants may set the feedback gains of their control policy to a moderate level across all trials as a compromise between expected success and expected effort on any given trial; that is, both hypotheses make the same prediction regarding the observed behavior.

line 254: must HAVE been

Fixed.

Figure 5: the dotted line seems to be the wrong color in the legend

Fixed.